# Low Serum Levels of Soluble Receptor Activator of Nuclear Factor κ B Ligand (sRANKL) Are Associated with Metabolic Dysregulation and Predict Long-Term Mortality in Critically Ill Patients

**DOI:** 10.3390/diagnostics12010062

**Published:** 2021-12-28

**Authors:** Tobias Puengel, Beate Weber, Theresa H. Wirtz, Lukas Buendgens, Sven H. Loosen, Lukas Geisler, Burcin Özdirik, Hamesch Karim, Samira Abu Jhaisha, Jonathan F. Brozat, Philipp Hohlstein, Albrecht Eisert, Eray Yagmur, Christian Trautwein, Frank Tacke, Alexander Koch

**Affiliations:** 1Department of Hepatology and Gastroenterology, Campus Virchow-Klinikum (CVK) and Campus Charité Mitte (CCM), Charité-Universitätsmedizin Berlin, 13353 Berlin, Germany; Lukas.geisler@charite.de (L.G.); burcin.oezdirik@charite.de (B.Ö.); frank.tacke@charite.de (F.T.); 2Department of Medicine III, RWTH-University Hospital Aachen, 52074 Aachen, Germany; beate.weber@rwth-aachen.de (B.W.); thwirtz@ukaachen.de (T.H.W.); lbuendgens@ukaachen.de (L.B.); khamesch@ukaachen.de (H.K.); sabujhaisha@ukaachen.de (S.A.J.); jbrozat@ukaachen.de (J.F.B.); phohlstein@ukaachen.de (P.H.); ctrautwein@ukaachen.de (C.T.); akoch@ukaachen.de (A.K.); 3Clinic for Gastroenterology, Hepatology and Infectious Diseases, University Hospital Düsseldorf, Medical Faculty of Heinrich Heine University Düsseldorf, 40225 Düsseldorf, Germany; Sven.Loosen@med.uni-duesseldorf.de; 4Hospital Pharmacy, RWTH-University Hospital Aachen, 52074 Aachen, Germany; aeisert@ukaachen.de; 5Institute of Clinical Pharmacology, RWTH-University Hospital Aachen, 52074 Aachen, Germany; 6Institute of Laboratory Medicine, Western Palatinate Hospital, 67655 Kaiserslautern, Germany; eyagmur@westpfalz-klinikum.de

**Keywords:** soluble receptor activator of nuclear factor κ B ligand (sRANKL), intensive care unit (ICU), critical illness, diabetes, glucose metabolism, adipokine, sepsis, inflammation, prognosis

## Abstract

Soluble receptor activator of nuclear factor κ B ligand (sRANKL) is a member of the tumor necrosis factor receptor superfamily, and therefore, involved in various inflammatory processes. The role of sRANKL in the course of bone remodeling via activation of osteoclasts as well as chronic disease progression has been described extensively. However, the potential functional importance of sRANKL in critically ill or septic patients remained unknown. Therefore, we measured sRANKL serum concentrations in 303 critically ill patients, including 203 patients with sepsis and 100 with non-sepsis critical illness. Results were compared to 99 healthy controls. Strikingly, in critically ill patients sRANKL serum levels were significantly decreased at intensive care unit (ICU) admission (*p* = 0.011) without differences between sepsis and non-sepsis patients. Inline, sRANKL was correlated with markers of metabolic dysregulation, such as pre-existing diabetes and various adipokines (e.g., adiponectin, leptin receptor). Importantly, overall mortality of critically ill patients in a three-year follow-up was significantly associated with decreased sRANKL serum concentrations at ICU admission (*p* = 0.038). Therefore, our study suggests sRANKL as a biomarker in critically ill patients which is associated with poor prognosis and overall survival beyond ICU stay.

## 1. Introduction

Among critically ill patients sepsis and septic shock still represent the most common cause of death, due to multiple organ failures as a result of dysregulated immune responses to an infection [1]. Despite great advances in therapy for critically ill patients early diagnosis and risk evaluation remain difficult but tremendously crucial for stratification according to prognosis and adequate management [2,3]. Therefore, biomarkers may ease and objectify early diagnosis besides clinical evaluation and common assessment scores [4]. Receptor activator of nuclear factor κ B (RANK) is a member of the tumor necrosis factor receptor family. Signaling of RANK is induced by RANK-ligand (RANKL) or its soluble form (sRANKL). Most comprehensive data of the RANK/RANKL system describe bone remodeling and repair via differentiation and activation of osteoclasts in combination with osteoprotegerin (OPG), a decoy receptor for RANKL [5,6]. Major intracellular downstream targets of RANK are nuclear factor κ B (NF-κB) and c-Jun N-terminal kinases (JNK), key regulators of inflammatory processes, antiviral responses and apoptosis [7]. Estrogen is a natural suppressor of RANKL explaining high rates of osteoporosis in postmenopausal females as activation of osteoclasts is suppressed ineffectively and bone turnover is stimulated [7]. Experimental work further demonstrated impaired lymph node development in RANKL deficient mice accentuating the interaction between bone metabolism and immunity as osteo-immunity [8,9]. Previous work highlighted the role of RANKL in adaptive immunity as RANKL is expressed on various immune cells, mainly T cells [10]. Moreover, recent studies elucidated the functional role of the RANKL/RANK/OPG in the central nervous system participating in thermoregulation and incorporating a protective role in ischemic stroke [11,12]. Kiechl et al. demonstrate that high sRANKL plasma concentrations are associated with type 2 diabetes mellitus (T2DM). Inline, blocking RANKL signaling in a mouse model of T2DM lowered glucose serum levels and improved insulin sensitivity [13]. Recent studies highlighted the relevance and described the potential roles of various growth factors and adipokines (e.g., Adiponectin, Leptin, Omentin and Resistin) in the course of critical illness and sepsis [14]. Adipokines are secreted from the adipose tissue and are, therefore, upregulated in obese patients displaying important regulators of body homeostasis, as well as immune responses to inflammation. sRANKL and its downstream targets are key regulators of a great spectrum of inflammatory responses, immune reactions and metabolic-related disorders, and therefore, potentially linked to critical illness. However, the relevance of sRANKL in critically ill and septic patients has not been reported to date and the functional role remains uncertain. We, therefore, investigated the clinical and diagnostic relevance of sRANKL serum concentrations as a potential biomarker in a large cohort of 303 critically ill patients.

## 2. Materials and Methods

### 2.1. Study Design

Our study cohort covered critically ill adults (*n* = 303) who were included at admission to our intensive care unit (ICU) at RWTH University Hospital Aachen from a consecutively recruiting, prospective observational trial. Written informed consent was obtained from the patient directly, his or her spouse, or the appointed legal guardian. Sepsis and septic shock (sepsis-3) was defined post hoc based on the third international consensus definition [1]. All other patients were classified as non-sepsis patients. The Berlin definition of acute respiratory distress syndrome (ARDS) was used to identify and classify patients with ARDS [15]. The patients, the patients’ relatives, or their primary care physicians were contacted to assess the long-term course. Intensive care treatments shorter than three days (e.g., due to post-interventional observation or intoxication) led to exclusion of the patient [16]; 99 healthy blood donors without any severe acute or chronic disease who were medically examined on a regular basis served as controls based on normal values for blood counts, C-reactive protein, and liver enzymes. The study protocol was approved by the local ethics committee and conducted in accordance with the ethical standards laid down in the 1964 Declaration of Helsinki (ethics committee of the University Hospital Aachen, RWTH-University, Aachen, Germany, reference number EK 150/06, approved on 2 November 2006).

### 2.2. sRANKL Measurements

Directly at the time of admission to ICU and prior to any therapeutic interventions blood samples were collected from the patients. Whole blood was centrifuged at 4 °C for 10 min, and serum aliquots of 1 mL were collected and frozen immediately at −80 °C. sRANKL serum concentrations were measured using a commercially available fluorescent immunoassay (Biomedica GmbH, Vienna, Austria) according to the manufacturer′s protocol by a scientist blinded to any clinical or other laboratory data of the patients or controls.

### 2.3. Statistical Analysis

Based to the skewed distribution of the parameters most data are demonstrated as median and range. Mann–Whitney U or chi-squared test were conducted for two-group analysis while Kruskal–Wallis test was used for analysis between multiple groups. Subgroup analysis was illustrated as box plots including median, quartiles, range, and extreme values of the given data with whiskers ranging from the minimum to the maximum value. Statistic outliers defined as a value that is smaller than the lower quartile minus 1.5 times the interquartile range or larger than the upper quartile plus 1.5 times the interquartile range were displayed as separate points [17]. Spearman linear correlation analysis was performed to assess differences between various variables. Kaplan–Meier curves were generated to estimate the survival function and a log-rank test was performed to assess differences between groups [18]. Receiver operating characteristic (ROC) curves were generated by plotting sensitivity against 1-specificity evaluating the value of a predictive marker or a composite score [19]. Differences between ROC curves were analyzed as previously described [20]. SPSS Version 23 (SPSS, Chicago, IL, USA) and MedCalc Version 16 (MedCalc Software, Ostend, Belgium) were used for all statistical analyses.

## 3. Results

### 3.1. sRANKL Serum Levels Are Reduced in Critically Ill Patients at ICU Admission

In critically ill patients various cytokines are elevated in the blood circulation due to organ failure and disrupted metabolism. As such sRANKL is a protein and member of the tumor necrosis family that can be detected in the bloodstream. Therefore, we measured sRANKL serum concentrations in 303 critically ill patients at admission to ICU before starting any specific interventions in comparison to 99 healthy control patients. Among all critically ill patients, we divided non-septic from septic patients including severe sepsis and septic shock. The complete study cohort includes 185 males and 118 females at a median age of 63 years at ICU admission ranging from 18 to 89 years (Table 1). Among all septic patients infection foci were identified: pulmonary (*n* = 115), abdominal (*n* = 33), urogenital (*n* = 8) or other (*n* = 47). Whereas, in non-septic patients the main cause of ICU admission was a cardiopulmonary event (*n* = 35) followed by decompensated liver cirrhosis (*n* = 17), acute pancreatitis (*n* = 13), severe gastrointestinal hemorrhage (*n* = 7), acute liver failure (*n* = 4) and other non-septic diseases (*n* = 24) (Table 2). Expectedly, sepsis patients had a significantly higher APACHE-II or SOFA score and revealed a higher proportion of mechanical ventilation as well as longer treatment duration on the ICU and increased 30-day and overall mortality (Table 1). Of interest, at the day of ICU admission sRANKL serum concentrations were significantly reduced in critically ill patients (median: 0.04, CI: 0–1.49) compared to healthy controls (median: 0.12, CI: 0–1.51) (*p* = 0.011). Additionally, further subdivision of all critically ill patients revealed decreased sRANKL serum concentrations in sepsis patients (median 0.04, CI: 0–1.49) compared to non-septic patients (median: 0.07, CI 0–1.39) (*p* = 0.085) (Figure 1).

### 3.2. sRANKL Plasma Concentrations Are Associated with Pre-Existing Diabetes and Metabolism-Related Serum Markers in Critically Ill Patients

To investigate the relevance of sRANKL in the course of disease in critically ill patients we performed extensive correlation analyses between sRANKL serum levels and various laboratory parameters, clinical scores as well as new or experimental biomarkers (Table 3, Figure 2). Our data indicate that sRANKL serum levels were not significantly associated with markers of inflammation (e.g., CRP, PCT) or clinical scores (e.g., APACHE-II, SOFA). Strikingly, sRANKL positively correlated with markers of organ failure or dysfunction, especially with markers of liver injury (e.g., ALT, prothrombin time) and cholestasis (e.g., yGT, ALP, bilirubin). Of note, diagnosis of type 2 diabetes before admission to ICU was associated with reduced sRANKL plasma concentrations (no diabetes median: 0.09, CI: 0–1.51, diabetes median: 0.01, CI: 0–1.49; *p* = 0.006) (Figure 3). Inline, sRANKL was correlated with glucose intolerance based on blood glucose levels (r = 0.174, *p* = 0.003) at ICU admission but we did not detect differences regarding long-term markers of insulin resistance (HbA1c, insulin) or dyslipidemia (cholesterol). Further features of the metabolic syndrome, such as obesity based on the BMI and lipometabolic disorders reflected by triglycerides, LDL cholesterol and total cholesterol were not significantly correlated with sRANKL serum levels. However, among a broad range of adipokines sRANKL positively correlated with adiponectin (r = 0.2, *p* = 0.03), leptin receptor (r = 0.232, *p* = 0.009) but not leptin itself and correlated inversely with ghrelin (r = −0.242, *p* = 0.01). Of note, sRANKL demonstrates a positive correlation with the human growth hormone (STH)—a relevant marker of various metabolic processes (e.g., lipolysis in adipose tissue, glucose metabolism, secretion of insulin) (Appendix A).

### 3.3. Decreased sRANKL Serum Levels at ICU Admission Are an Independent Predictor of Poor Overall Survival after Critical Illness

In critically ill patients various physiological functions are affected linked to inflammatory responses and metabolic disorders. As previously described by our group adipocytokines are associated with disease severity as well as short- and long-term mortality in critically ill patients [14]. From a total of 303 critically ill patients 226 patients initially survived their ICU stay. Among those patients, we could not observe a correlation between sRANKL plasma concentrations and ICU survival. In this study, we were able to assess not only short- but also long-term survival in a course of approximately three years after discharge from ICU, as the mortality risk of critically ill patients is significantly elevated beyond the ICU stay [21]. Strikingly, poor overall survival in long-term follow-up of three years was associated with decreased sRANKL levels at ICU admission (Figure 4A). Operating the Youden index we then determined the best cut-off value of sRANKL plasma concentrations as 0.015 pmol/L regarding sensitivity and specificity. Kaplan–Meier curve analysis revealed an association of low sRANKL serum levels at ICD admission with a poor long-term survival during three years of follow-up (survival median: 0.08, CI: 0–1.49; death median 0.02, CI: 0–1.32; *p* = 0.038) (Figure 4B). Performing Cox-regression analysis we further elucidated the prognostic value of sRANKL and exclude potential cofounders. Univariate Cox-regression analysis below the ideal cut-off value including various clinicopathological as well as laboratory markers of inflammation and organ dysfunction demonstrate that sRANKL serum concentrations at ICU admission turned out as a significant prognostic factor for poor long-term survival after critical illness (HR: 0.678, CI: 0.461–0.996, *p* = 0.048, Table 4). Using an optimal predictive cut-off value for the long-term survival of sRANKL at admission to ICU (0.015 pmol/L), ROC curve analysis revealed an AUC value of 0.567 regarding discrimination between 3 years survival and 3 years death (Figure 4C).

## 4. Discussion

In this study, we demonstrate that sRANKL plasma concentrations were significantly reduced in critically ill patients at ICU admission. Additionally, our data indicate that low sRANKL serum levels are an indicator of poor overall survival during long-term follow-up of 3 years after discharge from ICU. A potential explanation of this outcome prediction might be associated with the fact that sRANKL serum levels also correlated with various growth factors (e.g., ghrelin, somatotropin) and adipokines (e.g., leptin receptor, adiponectin) in our patient cohort. Although we did not observe a direct correlation of sRANKL with features of the metabolic syndrome, such as obesity and dyslipidemia, patients who were diagnosed with T2DM before admission to ICU had significantly lower levels of sRANKL compared to patients without a history of diabetes. Of note, a cross-sectional study including 40 prediabetic and 40 control subjects gave evidence for a positive correlation of sRANKL with prediabetes, BMI, insulin resistance (HOMA-IR) and also markers of inflammation (hs-CRP) [22]. Inline, another prospective study described a correlation of high sRANKL serum concentrations as a risk indicator for the development of T2DM. The authors were able to ameliorate glucose intolerance by suppressing sRANKL in two independent mouse models hypothesizing that RANKL is a potent activator of NF- κB on hepatocytes promoting cytokine expression and hepatic insulin resistance [13]. Kurihara et al. could furthermore demonstrate that sRANKL increased glucose uptake of murine macrophages in vitro by stimulating membrane translocation of Glut-1 [23]. Another study could demonstrate that features of the metabolic syndrome (waist circumference and dyslipidemia) significantly correlated with sRANKL serum levels in a pediatric population aged between 6 and 12 years [24]. In our study, we provide evidence for a positive correlation of sRANKL plasma levels with adiponectin, leptin receptor and ghrelin hypothesizing metabolic relevance of sRANKL in the course of critical illness. Adipokines as well as sRANKL itself recently gain growing research interest highlighting their functional potential beyond direct metabolic effects in the modulation of inflammatory and infectious disease [14,25]. For instance, adiponectin reflects a rather protective role in metabolic homeostasis and experimental models of sepsis linked anti-inflammatory responses to adiponectin [26,27]. In a former study of our group, we could demonstrate that leptin receptor serum levels were associated with parameters of inflammation (e.g., PCT) and cholestasis indicating poor ICU and overall survival [28]. On the contrary, ghrelin plasma concentrations are elevated in critically ill patients, correlating with markers of glucose metabolism in non-septic patients and predicting ICU survival [19]. Measuring circulating adipokines bear the potential to identify critically ill and septic patients as well as stratify the risk of severe disease progression and mortality. In this reference, sRANKL possibly complements those novel assessment tools of biomarkers. In this context, we have to discuss the interpretation of our data critically and disclose the limitations of our study design transparently. We present a large but monocentric study cohort and measurement of sRANKL serum levels has been conducted at one time point only during admission to ICU. Of note, it would be highly interesting for future studies to measure sRANKL serum concentrations not only on the day of ICU admission but also during the time of follow-up to better elucidate the value of sRANKL as a prognostic biomarker including even larger study cohorts. Measurements of sRANKL could also be integrated into existing scoring systems potentially optimizing risk stratification and prediction of overall mortality in critically ill patients. 

The functional implication of the RANK/RANKL system has widely been investigated in various tumor entities. In line with our work, most studies demonstrate a negative correlation between RANK expression and long-term survival as well as response to preoperative chemotherapy [29,30]. On the other hand, expression of RANK/RANKL also bears the potential of a therapeutic drug target. Pharmacologic inhibition of RANKL improved the efficacy of anti-CTLA-4 monoclonal antibodies in the therapy of solid tumors and experimental metastases [31,32,33,34]. One of the pharmacologic compounds is Denosumab-a human monoclonal antibody with a high binding affinity to RANKL leading to inhibition of RANK signaling [35]. Kaneko et al. analyzed the effects of glucocorticoid therapy on sRANKL serum levels in patients with autoimmune disease. The authors demonstrate that on average sRANKL plasma concentrations were unaltered over time—however, bone mineral density significantly decreased in those patients who initially had lower sRANKL serum levels before glucocorticoid therapy induction while bone mineral density even increased in patients with higher sRANKL concentration [36]. In reference to our study in critically ill patients, one could hypothesize that individual body composition might be associated with impaired long-term survival as initially low sRANKL levels might also affect bone metabolism and remodeling. Recent studies from our group highlighted the importance of the individual’s body composition reporting that excessive visceral adipose tissue, as well as sarcopenia and myosteatosis, are associated with poorer outcomes in critically ill patients [37,38,39], going in line with further studies discussing frailty as potential key determinant predicting complicated courses of disease and poorer long-term outcomes [40]. Numerous further functional pathways have been and are currently under investigation. For instance, an experimental study demonstrated that RANKL increased calcification of vascular smooth muscle cells by stimulating bone morphogenetic protein (BMP)4 expression hypothesizing relevance in the development of atherosclerosis in patients [41]. Inline, a recent and still ongoing study in women with postmenopausal osteoporosis reported that long-term application (24 months) of Denosumab had no effects on serum markers of atherosclerosis [42]. Inline, a monocentric study of a total of 472 patients revealed an association of coronary heart disease with decreased levels of sRANKL [43].

Based on our data, we hypothesize that sRANKL potentially has various metabolically relevant functions beyond bone remodelings, such as body homeostasis or glucose metabolism. Critical illness and sepsis possibly alter and affect those essential functions of the RANK/RANKL system which are then associated with impaired long-term overall survival after an ICU stay. In conclusion of our study, one could hypothesize that substitution of sRANKL bears the potential to influence the course of critical illness and sepsis beneficially. However, the long-term effects of a pharmacologic application with such a broad range of metabolic and immunologic pathways would require further in-depth analyses. Thus, the detailed functions of the RANK/RANKL system remain to be further investigated, representing an interesting target for future mechanistic and clinical studies.

## Figures and Tables

**Figure 1 diagnostics-12-00062-f001:**
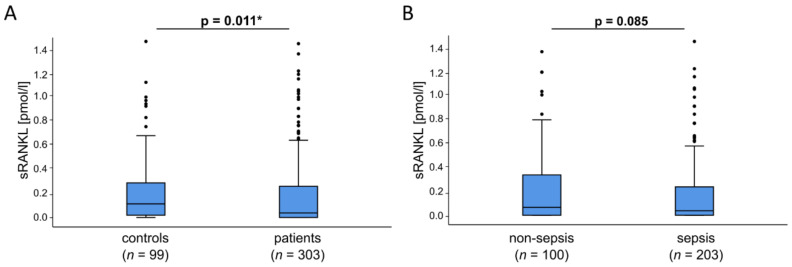
Serum sRANKL concentrations in critically ill patients and sepsis. (**A**) On the day of ICU admission, sRANKL serum concentrations were significantly reduced in critically ill patients compared to healthy controls (*p* = 0.011). (**B**) Among critically ill patients sRANKL serum levels showed reduced levels in sepsis patients compared to non-septic patients (*p* = 0.085). All data are shown as available. * *p* < 0.05.

**Figure 2 diagnostics-12-00062-f002:**
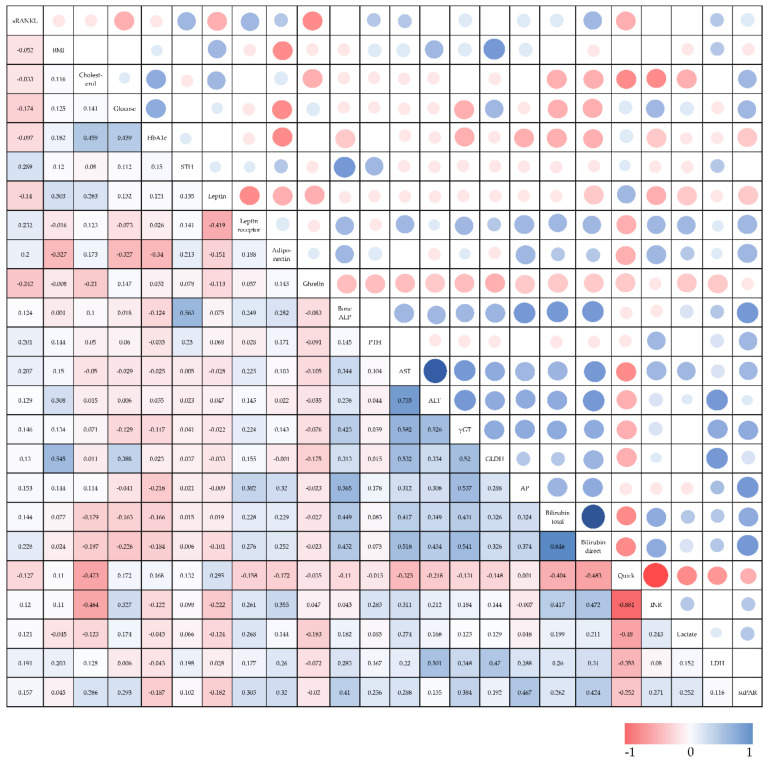
Correlation analysis of sRANKL serum levels with selected baseline characteristics and laboratory markers at ICU admission day as well as between multiple variables. Spearman rank correlation test was used to test significance. Heatmap presentation of positive or negative correlation. Abbreviations: BMI, body mass index; HDL, high density lipoprotein; LDL, low density lipoprotein; HGH, human growth hormone; ALP, alkaline phosphatase; PTH, parathyroid hormone; CRP, C-reactive protein; IL-6, interleukin 6; TNF-α, tumor necrosis factor α; AST, aspartate aminotransferase; ALT, alanine aminotransferase; γGT, gamma-glutamyl transpeptidase; GLDH, glutamate dehydrogenase; AP, Alkaline phosphatase; INR, International Normalized Ratio; LDH, lactate dehydrogenase; BNP, brain natriuretic peptide; GFR, glomerular filtration rate; APACHE, Acute Physiology and Chronic Health Evaluation; SOFA, sequential organ failure assessment; SAPS 2, Simplified Acute Physiology Score 2; suPAR, soluble urokinase plasminogen activator receptor.

**Figure 3 diagnostics-12-00062-f003:**
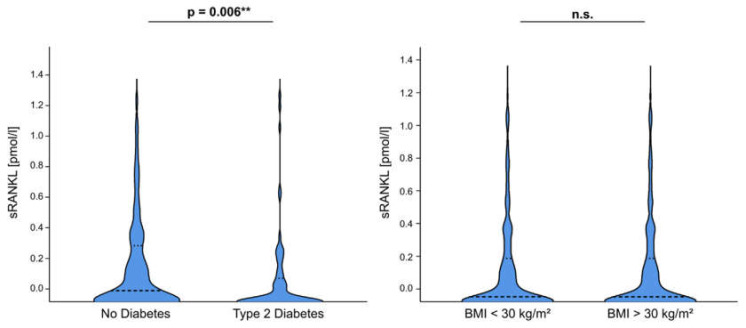
sRANKL serum levels are associated with type 2 diabetes. sRANKL serum concentrations were significantly decreased in patients diagnosed with type 2 diabetes but did not correlate with BMI classification (BMI < or > 30 kg/m^2^). All data are shown as available. ** *p* < 0.005, n.s.: not significant.

**Figure 4 diagnostics-12-00062-f004:**
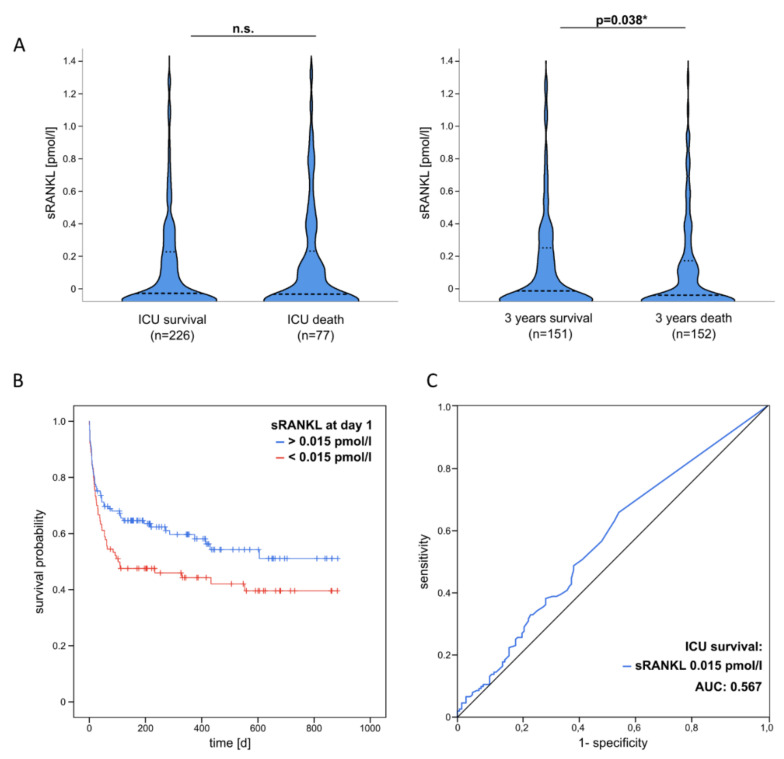
Prediction of ICU and long-term mortality by sRANKL serum levels. (**A**) sRANKL serum levels at ICU admission did not correlate with ICU survival but predict a poor overall survival in the long-term (three years of follow-up). (**B**) Kaplan–Meier survival curves of critically ill patients are displayed for the three years of follow-up demonstrating that patients with sRANKL serum levels below a cut-off value of 0.015 pmol/L had a higher probability of long-term mortality. (**C**) ROC curve analysis reveals that sRANKL serum concentrations at an optimal predictive cut-off value for long-term survival (0.015 pmol/L) at admission to ICU had an AUC value of 0.567 regarding discrimination between 3 years survival and 3 years death. All data are shown as available. * *p* < 0.05, n.s.: not significant.

**Table 1 diagnostics-12-00062-t001:** Baseline patient characteristics and Soluble receptor activator of nuclear factor κ B ligand (sRANKL) plasma measurements.

Parameter	All Patients	Sepsis	Non-Sepsis	*p* *
Number	303	203	100	
Sex (male/female)	185/118	124/79	61/39	n.s.
Age median, (range in years)	63 (18–89)	65 (21–89)	60 (18–85)	n.s.
BMI median, (range in kg/m^2^)	26 (16–87)	26 (17–87)	26 (16–53)	n.s.
Type 2 Diabetes presence (%)	82 (27)	54 (27)	28 (28)	n.s.
APACHE-II score, median (range)	18 (2–43)	19 (3–43)	15 (2–33)	<0.001
SOFA score, median (range)	9 (0–19)	10 (0–19)	7 (0–17)	0.001
SAPS 2 score, median (range)	43 (0–80)	44 (0–79)	42 (13–80)	n.s.
Mechanical ventilation, *n* (%)	226 (75)	160 (79)	66 (66)	0.016
Vasopressor demand, *n* (%)	167 (55)	123 (60)	44 (44)	<0.001
ICU days, median (range)	9 (1–357)	12 (1–137)	7 (1–357)	<0.001
30-day mortality, *n* (%)	95 (31)	67 (33)	28 (28)	<0.001
3-year mortality, *n* (%)	152 (50)	88 (43)	64 (64)	<0.001
sRANKL, median (range in pmol/L)	0.04 (0–2.58)	0.04 (0–2.58)	0.07 (0–1.79)	0.085
CRP, median (range in mg/L)	93 (5–230)	152 (5–230)	17 (5–230)	<0.001
Leucocytes, median (range in per nL)	12.5 (1.8–149.0)	12.9 (1.8–149.0)	12.1 (1.8–29.6)	0.025
Cystatin C, median (range in mg/L)	1.81 (0.41–7.57)	2.00 (0.41–7.57)	1.26 (0.41–5.41)	<0.001
Bilirubin, median (range in mg/dL)	0.8 (0.1–40.4)	0.78 (0.1–40.4)	0.8 (0.1–39.1)	n.s.

For quantitative variables, median and range (in parenthesis) are given. * Significance between sepsis and non-sepsis patients was assessed using the Mann–Whitney U test or chi-squared test. Abbreviations: BMI, body mass index; APACHE, Acute Physiology and Chronic Health Evaluation; SOFA, sequential organ failure assessment; SAPS 2, Simplified Acute Physiology Score 2; ICU, intensive care unit; sRANKL, Soluble receptor activator of nuclear factor κ B ligand; CRP, C-reactive protein.

**Table 2 diagnostics-12-00062-t002:** Disease etiology of the study population leading to ICU admission.

Etiology in Critically Ill Patients	Sepsis *n* = 203
Pulmonary (%)	115 (57)
Abdominal (%)	33 (16)
Urogenital (%)	8 (4)
Other (%)	47 (23)
**Etiology of non-sepsis in critically ill patients**	**Non-Sepsis** ***n* = 100**
Cardio-pulmonary disorder (%)	35 (35)
Decompensated liver cirrhosis (%)	17 (17)
Acute pancreatitis (%)	13 (13)
Severe gastrointestinal hemorrhage (%)	7 (7)
Acute liver failure (%)	4 (4)
Other (%)	24 (24)

Data are shown in absolute numbers and % (in parenthesis).

**Table 3 diagnostics-12-00062-t003:** Correlations of sRANKL with baseline characteristics, laboratory markers and clinical scores at ICU admission day.

ICU Patients			
	r	*p*	
Obesity and diabetes			
BMI	−0.052	0.385	
Triglycerides	−0.006	0.926	
Cholesterol	−0.033	0.605	
HDL	−0.092	0.317	
LDL	−0.056	0.543	
Glucose	−0.174 **	0.003 **	
HbA1c	−0.097	0.282	
Insulin	−0.054	0.553	
C−Peptide	0.008	0.926	
STH	0.289 **	0.002 **	
Leptin	−0.140	0.119	
Leptin receptor	0.232 **	0.009 **	
Adiponectin	0.2 *	0.03 *	
Ghrelin	−0.242 **	0.01 *	
Bone metabolism			
Bone ALP	0.124	0.208	
PTH	0.201 *	0.027 *	
Vitamin D3	−0.009	0.927	
Serum calcium	0.01	0.859	
Markers of inflammation			
CRP	−0.022	0.707	
Procalcitonin	0.076	0.262	
IL6	0.087	0.186	
TNFα	0.209 *	0.047 *	
Liver injury and cholestasis			
AST	0.207 **	<0.001 ***	
ALT	0.129 *	0.026 *	
γGT	0.146 *	0.011 *	
GLDH	0.13 *	0.032 *	
AP	0.153 *	0.01 *	
Bilirubin total	0.144 *	0.012 *	
Bilirubin direct	0.228 **	0.001 **	
Prothrombin time	0.085	0.144	
Quick	−0.127 *	0.028 *	
INR	0.12 *	0.04 *	
Albumin	−0.078	0.304	
Urea	−0.023	0.695	
Lactate	0.121 *	0.039 *	
LDH	0.191 *	<0.001 ***	
NTproBNP	−0.043	0.596	
Renal function			
Cystatin C	0.022	0.764	
GFR	0.043	0.533	
Clinical scores			
APACHE−II	0.008	0.897	
SOFA	0.054	0.538	
SAPS 2	−0.085	0.38	
New and experimental biomarkers			
suPAR	0.157 *	0.022 *	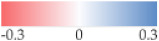

Spearman rank correlation test was used to test significance; the Spearman’s rho correlation coefficient is depicted as “r” with * *p* < 0.05; ** *p* < 0.01; *** *p* < 0.001. Heatmap presentation of positive or negative correlation. Abbreviations: BMI, body mass index; HDL, high density lipoprotein; LDL, low density lipoprotein; HGH, human growth hormone; ALP, alkaline phosphatase; PTH, parathyroid hormone; CRP, C-reactive protein; IL-6, interleukin 6; TNF-α, tumor necrosis factor α; AST, aspartate aminotransferase; ALT, alanine aminotransferase; γGT, gamma-glutamyl transpeptidase; GLDH, glutamate dehydrogenase; AP, Alkaline phosphatase; INR, International Normalized Ratio; LDH, lactate dehydrogenase; BNP, brain natriuretic peptide; GFR, glomerular filtration rate; APACHE, Acute Physiology and Chronic Health Evaluation; SOFA, sequential organ failure assessment; SAPS 2, Simplified Acute Physiology Score 2; suPAR, soluble urokinase plasminogen activator receptor.

**Table 4 diagnostics-12-00062-t004:** Univariate Cox-regression analysis of sRANKL serum concentrations in critically ill patients.

Parameter	Univariate Cox Regression	
	*p*-Value	Hazard Ratio (95% CI)
sRANKL	0.048 *	0.678 (0.461–0.996)
Age	0.002 **	1.022 (1.008–1.036)
Sex	0.398	1.185 (0.800–1.754)
BMI	0.045 *	0.965 (0.931–0.999)
Leukocytes	0.340	0.989 (0.967–1.012)
CRP	0.314	1.001 (0.999–1.003)
Bilirubin total	0.005 **	1.057 (1.026–1.088)
Creatinine	0.532	1.020 (0.959–1.085)
Hemoglobin	0.029 *	0.991 (0.982–0.999)

* *p* < 0.05; ** *p* < 0.01. Abbreviations: BMI, body mass index; CRP, C-reactive protein.

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
