# Peer review of "Low Serum Levels of Soluble Receptor Activator of Nuclear Factor κ B Ligand (sRANKL) Are Associated with Metabolic Dysregulation and Predict Long-Term Mortality in Critically Ill Patients"

_diagnostics, 2021, doi:10.3390/diagnostics12010062_

Round 1

Reviewer 1 Report

Many thanks for the opportunity to review this interesting manuscript which tries to elucidate the role of a new biomarker in critical illness.

The authors demonstrated, that reduced sRANKL serum levels are associated with worse outcome. I have a few suggestions to the Authors, which may help to strengthen the manuscript.

  1. As the data distribution is skewed, it would be better to present the datapoints as violin-plots instead of box and whiskers to emphasise the spread.
  2. The Table and the Figures with the correlations is quite difficult to read and interpret. I'd either suggest a heatmap type graphical representation or just to highlight the significant correlations in the main manuscript and move the individual figures to the supplement.
  3. I'd urge the authors NOT to say something is "trending towards xyz" Either there is a statistically significant and clinically relevant difference or there isn't. PLease revise these throughout the manuscript.
  4. There is a striking reversal in the short vs longer term mortality between the sepsis and non-sepsis groups. This could well be a result of progression of multimorbidity (liver disease in particular) or that the non-sepsis groups were more frail. There is a wealth of literature showing that frailty is associated with poorer long-term outcomes and this might be an interesting angle to discuss (one example is https://www.nature.com/articles/s41598-021-92874-w ,however there are plenty of other papers around). As sRANKL seems to be implicated in bone metabolism and frailty can be described as in part a metabolic disorder, I wonder if there is a link between them. Could the authors determine the HFRS or eFI scores for their patients? I'd understand if they can't however with their electronic database it is possible that they can gather the necessary ICD10 codes for these markers.
  5. With the current data, the Authors didn't demonstrate that sRANKL is predictive of worse outcomes. To do that they would need to analyse their data differently and create a relatively complex regression analysis. I'd argue that they have demonstrated that sRANKL is associated (not correlated!) with worse outcomes. It would be important to revise the paper and clear up the inconsistencies of language, unless they can support it with appropriate statistical analysis.
  6. In the Discussion the text between L259 and L279 is largely redundant and not advancing the points the authors would like to make. Please consider removing this.

Author Response

Point-to-point Responses to the Reviewer’s Comments
Reviewer 1
Many thanks for the opportunity to review this interesting manuscript which tries to elucidate the role of a new biomarker in critical illness. The authors demonstrated that reduced sRANKL serum levels are associated with worse outcome. I have a few suggestions to the Authors, which may help to strengthen the manuscript.
Response: We thank the reviewer for his/her positive evaluation of our manuscript.
1. As the data distribution is skewed, it would be better to present the datapoints as violin-plots instead of box and whiskers to emphasise the spread.
Response: We thank the reviewer for his/her comment and adapted all figures as recommended, now showing our results as violin plots instead of box-and-whisker plots emphasizing the skewed data distribution.
2. The Table and the Figures with the correlations is quite difficult to read and interpret. I'd either suggest a heatmap type graphical representation or just to highlight the significant correlations in the main manuscript and move the individual figures to the supplement.
Response: We thank the reviewer for his/her comment and adjusted the figures and manuscript as recommended. We included a heatmap type graphical data representation to “Table 3” helping the reader to grasp the key messages at first sight and also highlighted significant results in the main manuscript. From our point of view, scatterplot data demonstration can help the reader to gain a better insight into single datapoint distribution increasing transparency of our results. As recommended by the reviewer, we excluded the scatterplot data demonstration from “Figure 2” and moved those results to a new supplementary “Figure S1” and integrated this new figure to the manuscript.
3. I'd urge the authors NOT to say something is "trending towards xyz" Either there is a statistically significant and clinically relevant difference or there isn't. PLease revise these throughout the manuscript.
Response: We thank the reviewer for his/her comment and adjusted the phrasing as recommended.
4. There is a striking reversal in the short vs longer term mortality between the sepsis and non-sepsis groups. This could well be a result of progression of multimorbidity (liver disease in particular) or that the non-sepsis groups were more frail. There is a wealth of literature showing that frailty is associated with poorer long-term outcomes and this might be an interesting angle to discuss (one example is https://www.nature.com/articles/s41598-021-92874-w ,however there are plenty of other papers around). As sRANKL seems to be implicated in bone metabolism and frailty can be described as in part a metabolic disorder, I wonder if there is a link between them. Could the authors determine the HFRS or eFI scores for their patients? I'd understand if they can't however with their electronic database it is possible that they can gather the necessary ICD10 codes for these markers.
Response: We would like to thank the reviewer for his/her expert comment mentioning a very important aspect in this field of research. We fully agree with the latest research results demonstrating that frailty is associated with poorer long-term survival. As the reviewer mentioned, the RANKL/RANK/OPG complex is intensively interconnected with factors potentially associated with frailty such as bone metabolism and consequently also with osteoporosis. Based on our data we can only speculate on a correlation of sRANKL serum levels and frailty (see now in L315-329 in the updated manuscript). Unfortunately, it is neither possible for us to provide a Hospital Frailty Risk Score (HFRS) nor the Electronic Frailty Index (eFI). As mentioned, we indeed have full access to all ICD10 codes, but assessment of a frailty score based on the existing data or the ICD10 codes requires prior internal validation and was not implemented in our routine coding practises during the recruitment phase of the cohort. Therefore, it is not possible from our side to provide profound data generating a frailty index. During the analysis of this study, we also tried to approach the reviewer´s question in line to a previous study from our group (“Skeletal Muscle Composition Predicts Outcome in Critically Ill Patients” PMID: 32832910) based on CT imaging and assessment of bone mineralisation and muscle or fat composition of our patients. However, this complex analysis is only possible if one can provide a high frequency of CT scans for a longer period of time. We can provide those data for a certain patient cohort such as patients diagnosed severe pancreatitis who have repetitive CT scans over time to assess development of necrotic areas for example. Therefore, this analysis would only be possible for a very small part of our study cohort. Nevertheless, this is an expert comment and we already started implementing assessment of frailty in our analyses for future studies, as frailty is a key parameter determining patients´ outcome. 

5. With the current data, the Authors didn't demonstrate that sRANKL is predictive of worse outcomes. To do that they would need to analyse their data differently and create a relatively complex regression analysis. I'd argue that they have demonstrated that sRANKL is associated (not correlated!) with worse outcomes. It would be important to revise the paper and clear up the inconsistencies of language, unless they can support it with appropriate statistical analysis.
Response: We thank the reviewer for his/her comment and adjusted the phrasing as recommended throughout the manuscript. In addition, we provide complex univariate and multivariate Cox-regression analyses demonstrating statistical data in the main text as well as a new “Table 4” further elucidating the prognostic value of sRANKL and excluding potential cofounders. Univariate Cox-regression analysis below the ideal cut-off value including various clinicopathological as well as laboratory markers of inflammation and organ dysfunction demonstrate that sRANKL serum concentrations at ICU admission turned out as a significant prognostic factor for poor long-term survival after critical illness. However, sRANKL did not withstand during multivariate Cox-regression analyses with a stepwise backward selection of variables including parameters with potential prognostic relevance in univariate Cox-regression analysis. Therefore, both reviewers raised an important aspect which is now adjusted in the result section and we adapted the phrasing throughout the manuscript.
6. In the Discussion the text between L259 and L279 is largely redundant and not advancing the points the authors would like to make. Please consider removing this.
Response: We would like to thank the reviewer his/her helpful suggestion. As recommended, we removed large parts of this section and apologise for any redundancy. In this section, we would like to compare our findings to existing study results (prognostic value of sRANKL in neoplastic diseases; already existing pharmacologic molecules (Denosumab) to potentially target RANKL) and highlight the clinical potential of sRANKL. We also tried to link the key messages we intended to deliver to the reader closer to our study, now.

Reviewer 2 Report

Review of:

Low serum levels of soluble receptor activator of nuclear factor κB ligand (sRANKL) are associated with metabolic dysregulation and predict long-term mortality in critically ill patients

1 - Please provide statistical results in the abstract, especially the association test between decreased sRANKL levels and three years mortality.

2 - Please do not mention severe sepsis in the introduction (1st sentence). In the Sepsis 3 definition, severe sepsis does not exist anymore. And it is well explained in the methods.

3 - Lines 133, 163 and 210, please provide the medians of sRANKL (pmol/L) in both groups and the confidence interval before the p value. These values are never written in the manuscript, not in the text nor in the legend of figure 1, whereas there are the main results of your study.

4 - Also provide the “r” values before the p values, lines 164, 170 and 171.

5 - What does “overall mortality” mean? We can understand that it is about 3 years mortality, please clarify this point in the table 1 and in the discussion.

6 - Please show ROC curve with sRANKL at admission at 0.015pmol/L to predict 3 years survival. Table 2 may seem less relevant than a ROC curve.

7 - You have many data regarding these 303 patients; it is unfortunate to only report univariate analysis. If the goal is to show that sRANKL dosing could ever be clinically helpful in predicting long-term survival of critically ill patients, multivariate analysis including at least patient’s age and comorbidities is expected. Without such analysis, sentence lines 153-154-155 cannot be true.

8 - The discussion should be more centred on ICU patients and biomarkers. Patients with very higher BMI have lower sRANKL levels, even if statistics are not significant, a comment about the obesity paradox in sepsis? Any ideas to try inhibiting RANK expression to improve sepsis outcomes?

Author Response

Point-to-point Responses to the Reviewer’s Comments
Reviewer 2
1. Please provide statistical results in the abstract, especially the association test between decreased sRANKL levels and three years mortality.
Response: We thank the reviewer for his/her comment and added statistical analyses to the abstract:
- sRANKL serum levels at intensive care unit (ICU) admission (p=0.011)
- association of sRANKL with three-year survival / mortality (p=0.038)
2. Please do not mention severe sepsis in the introduction (1st sentence). In the Sepsis 3 definition, severe sepsis does not exist anymore. And it is well explained in the methods.
Response: We thank the reviewer for his/her expert comment and apologise for this mistake. We discarded `severe sepsis` from the first sentence as mentioned according to `The Third International Consensus Definitions for Sepsis and Septic Shock`.
3. Lines 133, 163 and 210, please provide the medians of sRANKL (pmol/L) in both groups and the confidence interval before the p value. These values are never written in the manuscript, not in the text nor in the legend of figure 1, whereas there are the main results of your study.
Response: We thank the reviewer for his/her comment and adjusted the manuscript according to his/her recommendation. Thank you for helping to increase validity of our work.
4. Also provide the “r” values before the p values, lines 164, 170 and 171.
Response: We thank the reviewer for his/her comment and adjusted the manuscript accordingly.
5. What does “overall mortality” mean? We can understand that it is about 3 years mortality, please clarify this point in the table 1 and in the discussion.
Response: We thank the reviewer to help clarifying this important aspect. In our study we provide a three-year follow-up of our patients to analyse survival / mortality after discharge from the ICU. We now adjusted this important aspect throughout our manuscript.
6. Please show ROC curve with sRANKL at admission at 0.015pmol/L to predict 3 years survival. Table 2 may seem less relevant than a ROC curve.

Response: We thank the reviewer for his/her comment and added ROC curve analyses to the main manuscript and demonstrate results in “Figure 3” using an optimal predictive cut-off value for long-term survival of sRANKL at admission to ICU (0.015 pmol/l); AUC: 0.567.
7. You have many data regarding these 303 patients; it is unfortunate to only report univariate analysis. If the goal is to show that sRANKL dosing could ever be clinically helpful in predicting long-term survival of critically ill patients, multivariate analysis including at least patient’s age and comorbidities is expected. Without such analysis, sentence lines 153-154-155 cannot be true.
Response: We thank both reviewers for their expert comments raising this question independently from each other (see comment 5. From reviewer 1). As already answered above, we now provide complex univariate and multivariate Cox-regression analyses demonstrating statistical data in the main text as well as a new “Table 4” further elucidating the prognostic value of sRANKL and excluding potential cofounders. Univariate Cox-regression analysis below the ideal cut-off value including various clinicopathological as well as laboratory markers of inflammation and organ dysfunction demonstrate that sRANKL serum concentrations at ICU admission turned out as a significant prognostic factor for poor long-term survival after critical illness. However, sRANKL did not withstand during multivariate Cox-regression analyses with a stepwise backward selection of variables including parameters with potential prognostic relevance in univariate Cox-regression analysis. Therefore, both reviewers raised an important aspect which is now adjusted in the result section and we adapted the phrasing throughout the manuscript.
8. The discussion should be more centred on ICU patients and biomarkers. Patients with very higher BMI have lower sRANKL levels, even if statistics are not significant, a comment about the obesity paradox in sepsis? Any ideas to try inhibiting RANK expression to improve sepsis outcomes?
Response: We thank the reviewer for his/her expert comments. The referee is completely right that the “obesity paradox” is an important aspect to discuss. Recent studies and meta-analyses give evidence that higher BMIs were associated with better survival of critically ill patients at ICU (PMID: 27306751: Increased body mass index and adjusted mortality in ICU patients with sepsis or septic shock: a systematic review and meta-analysis). The authors discuss potential protective effects of obesity:
- increased renin-angiotensin system activity during obesity might decrease the need for fluids or vasopressor support
- increased lipoprotein levels and adipose tissue in obese patients might increase the volume of distribution relatively decreasing lipopolysaccharide levels or other harmful bacterial products
- beneficial energy storage during the catabolic septic state
In line, recent studies from our group could demonstrate that individual´s body composition is another important parameter affecting mortality reporting that excessive visceral adipose tissue as well as sarcopenia and myosteatosis are associated with poorer outcome in critically ill patients (CT-based determination of excessive visceral adipose tissue is associated with an impaired survival in critically ill patients PMID: 33861804; Skeletal Muscle Composition Predicts Outcome in Critically Ill Patients PMID: 32832910).
Regarding pharmacologic targeting the RANKL/RANK/OPG system we can only speculate on potential effects in the course of critical illness. At the current state of knowledge, we would urgently need further translational data and studies investigating RANKL/RANK in experimental models. In a next step, evaluating pharmacologic effects - for example, using an available inhibitor of RANKL such as Denosumab - might be highly interesting and lead to further analysis in clinical studies.
We thank the reviewer for his/her comments and integrated that discussion to the main manuscript (L292-328).

Round 2

Reviewer 1 Report

Thank you very much for addressing most of my comments in this much improved version. 

I note that the authors tried to use regression modelling to see if sRANKL levels are predictive of outcome. I'd advise against this, as their database is relatively small and they can't control for all the confoundings without producing a severely overfitting model. The univariate Cox regression analysis shows that there might be a statistically significant, but very weak association between sRANKL levels and mortality, however this would need to be adjusted for multiple factors. Selecting the appropriate variables to fit into a model where the event count and the sample size is small, without producing a model which is not generalisable is a difficult task and should be done with careful thought for advanced variable selection strategies. Indeed, the ROC analysis is fairly disappointing as an AUC of 0.567 is only marginally better than flipping a coin at ICU admission. It is also biologically difficult to understand, how a blood result taken at one timepoint early in the critical care course would be helpful in predicting survival after 3 years!

I'd very seriously consider removing this part of the analysis from the manuscript.

Apologies for not being clear with my heatmap comment! The new version is significantly better than the previous table, however I was looking for something like the figure below, which can provide greater clarity and would allow the authors to showcase the correlations between multiple variables:

Lastly, the authors didn't include any references to studies pointing at frailty being a significant determinant of outcome. In line 320 the new reference 39 is a meta-analysis looking at the effect of obesity on critical care survival. I presume the authors wanted to use this study to strengthen their case for the first part of the sentence?

Reviewer 2 Report

This revised version is suitable for reviewing, and seems adequate to publish.

This work opens up interesting research field.

Round 3

Reviewer 1 Report

Thanks for the responses.